# The Interplay between Enucleated Sieve Elements and Companion Cells

**DOI:** 10.3390/plants12173033

**Published:** 2023-08-23

**Authors:** Angel J. Matilla

**Affiliations:** Departamento de Biología Funcional, Universidad de Santiago de Compostela, 14971-Santiago de Compostela, Spain; angeljesus.matilla@usc.es

**Keywords:** sieve element, sap, enucleation, companion cell, endoplasmic reticulum, transcription factors, lipophilic compounds, plasmodesmata, P-proteins

## Abstract

In order to adapt to sessile life and terrestrial environments, vascular plants have developed highly sophisticated cells to transport photosynthetic products and developmental signals. Of these, two distinct cell types (i.e., the sieve element (SE) and companion cell) are arranged in precise positions, thus ensuring effective transport. During SE differentiation, most of the cellular components are heavily modified or even eliminated. This peculiar differentiation implies the selective disintegration of the nucleus (i.e., enucleation) and the loss of cellular translational capacity. However, some cellular components necessary for transport (e.g., plasmalemma) are retained and specific phloem proteins (P-proteins) appear. Likewise, MYB (i.e., *APL*) and NAC (i.e., *NAC45* and *NAC86*) transcription factors (TFs) and OCTOPUS proteins play a notable role in SE differentiation. The maturing SEs become heavily dependent on neighboring non-conducting companion cells, to which they are connected by plasmodesmata through which only 20–70 kDa compounds seem to be able to pass. The study of sieve tube proteins still has many gaps. However, the development of a protocol to isolate proteins that are free from any contaminating proteins has constituted an important advance. This review considers the very detailed current state of knowledge of both bound and soluble sap proteins, as well as the role played by the companion cells in their presence. Phloem proteins travel long distances by combining two modes: non-selective transport via bulk flow and selective regulated movement. One of the goals of this study is to discover how the protein content of the sieve tube is controlled. The majority of questions and approaches about the heterogeneity of phloem sap will be clarified once the morphology and physiology of the plasmodesmata have been investigated in depth. Finally, the retention of specific proteins inside an SE is an aspect that should not be forgotten.

## 1. Brief Background on Photosynthetic Organisms’ Adaptation to Land: The Key Role of Phloem

Land plants evolved from charophycean freshwater green algae around 450 million years ago [1,2]. However, the transition from the aquatic to the terrestrial environment is still being studied. These studies have demonstrated the great potential of liverwort *Marchantia polymorpha* as an excellent genetic model for studying the conserved and diversified mechanisms underlying the development of land plants [3]. Thus, the study of the *Marchantia polymorpha* genome has shown that, in comparison to its relative Charophycean (green algae), liverwort is characterized by innovative biochemical pathways, including—among others—the auxin signaling pathway [4]. This was possibly the first instance of hormonal signaling that triggered the differentiation of primitive transporting tissues by inducing an organized, programmed cell death (i.e., cellular-restricted emptying) [5]. Genome analysis of the *Marchantia polymorpha* revealed that the auxin signaling pathway was already developed in early nonvascular plants.

Adaptation to land, water uptake, and body support started during the transition of the nonvascular plants to terrestrial habitats. The establishment of plant life on land constituted one of the most outstanding evolutionary events in Earth’s history [6]. The first land plants, incapable of growing tall, appeared on the surface of the Earth 475 million years ago [7]. Later (i.e., 423 million years ago), land plants with the ability to grow in height appeared, demonstrating the ability to capture more light energy [8]. Taken together, sessile life and terrestrial colonization have been attributed to a series of major improvements in plant body plans, anatomy, and biochemistry, resulting in a substantial decrease in atmospheric CO_2_ levels and progressive oxygenation of the environment [9]. In order to acquire nutrients from the soil and remain anchored to it, primitive photosynthetic organisms acquired a plastic root system [10,11]. Over time, the root system rapidly extended its functionality and complexity [11]. Likewise, evolution produced land plants with a stomatal apparatus through which CO_2_ and water vapor exchanged between the atmosphere and leaf to regulate two vital physiological processes (i.e., photosynthesis and transpiration) [12].

## 2. Emergence of Vasculature in Plants

As a consequence of the above, plants have had to evolve by acquiring highly specialized vascular tissues which enable the long-distance translocation of essential molecules for their growth and survival (i.e., water, nutrients, signal molecules, and photoassimilates) [13,14]. Thus, soil and aerial parts must connect via a complex organ (i.e., stem) which, in addition to having cell walls (CWs) enriched in lignin between the cellulose matrix to keep the organism erect, possessed an interior composed of two highly specialized vasculature systems for long-distance transport: (i) the secondary xylem, also called “wood”, for the unidirectional apoplastic (upward) transport of liquid water and soil nutrients from roots to shoots and leaves via the transpiration stream [15] and (ii) the phloem, also called “bast”, for bidirectional and active simplastic distribution, mediated by plasmodesmata, of organic nutrients generated in the photosynthetic source organs in the presence of photosynthetically competent light [16]. The functions of xylem and phloem strongly depend on each other. Interestingly, auxin collaborates in the coordination of the determination of vascular precursors [17]. However, it is not entirely clear which cell types produce auxins. The thick and lignified CW of xylem cells provides mechanical support for the plant and acts as a notable barrier against pests and pathogens. In other words, lignin plays a pivotal role in vascular tissues and is considered one of the hallmarks of terrestrial plants; therefore, its appearance constituted a key step in the evolution of Spermatophytes [13,18,19].

The xylem vessels undergo a programmed cell death followed by autolysis, which results in the total clearance of protoplasmic contents (i.e., fully empty cells). However, the sieve tube connective cells (i.e., highly specialized sieve elements: SEs) are living structures at the time of maturity and are physiologically active due to an appropriate meristem differentiation. However, little is known about how the molecular genetic specification of phloem cells is controlled since, before the onset of the differentiation, SE precursor cells do not show any obvious cytological differences compared with neighboring cells [20]. That is, during its differentiation, the SE undergoes a selective and partial autolysis in which some organelles are lost, while others are retained and rearranged with drastic modifications. In other words, this process can be likened to a highly selective programmed cell death that is without parallel amongst plant cells. This peculiar differentiation implies the selective disintegration of the nucleus (enucleation) and nucleolus [21], vacuole, peroxisomes, Golgy stacks, and a good number of plastids. This disintegration generates the low hydraulics necessary for the unimpeded movement of the sap. Likewise, the cytosol also remains partially degraded (i.e., low density) with very small volume and a peripheral location. However, plasmalemma, reinforced CWs, phloem-specific proteins, some population of ribosomal fragments (i.e., molecular debris originated during SE maturation), functioning chloroplasts, and plastids are divided into two types (S-type plastids, which contain starch inclusions, and P-type plastids, which contain proteinaceous inclusion bodies and are typical of monocots) and other organelles are retained in a modified form [13,22]. During SE differentiation phloem protein (P-protein) bodies appear. In mature SE, P-proteins take the form of filaments, tubules, and crystalline or amorphous structures. Filaments of P-protein appear in Angiosperms sieve tubes aligned with plasmalemma or, alternatively, on the sieve plates. Likewise, the mature SE have small and scarce mitochondria that are more rudimentary than those found in other cell types, probably due to the low catabolic activity of SE. However, available evidence suggests that mitochondria remain present and metabolically active in mature SE (Figure 1). Do mitochondria play any role in mature SEs? This question cannot be answered because it is unknown how much energy the ES requires to be physiologically competent. On the other hand, in mature SEs, modified organelles such as the plastids and mitochondria are eventually aligned with the parietal CWs and the endoplasmic reticulum (ER) undergoes structural changes by pressing stacks of ribosomes-free membranes to the plasma membrane. Interestingly, the smooth ER cisternae are believed to act as intracellular Ca^2+^-sequestration compartments. The ER is peripheral and polymorphic, with a highly dynamic structure. In plant cells, its mobility depends on the actin–myosin cytoskeleton. This implies severely reduced ER mobility in SEs, which lack actin filaments. The plasmalemma, with a large number of carriers, channels, and pumps, is indispensable for long-distance transport from sources to sinks. Nevertheless, some of the SE components in the sieve tube have a role in the life of the differentiated phloem that has yet to be determined. Unlike P-proteins, membrane-bound organelles are restricted to a thin parietal layer in functional SEs [23]. Considering the above, the SEs in their mature state are no longer autonomous cells but become strongly dependent on the support of the neighboring companion cells. Finally, the formation of the sieve plate during the differentiation process is an important acquisition. As a whole, the sieve element structural integrity is highly sensitive to manipulations and, thus, establishing the approximate state of its structure is not always possible [24].

Currently, few genes have been proven to be directly involved in phloem development. However, some progress has been made in unraveling them in recent years [25]. Although studies at the molecular level of the differentiation of SE in the stem remain very scarce because the SE structural integrity is highly sensitive to manipulation, the knowledge of this process at the root level is somewhat more advanced [25]. This special differentiation process allows the sieve tube to transport molecules throughout their symplastic space. The transport occurs through sieve tube elements, narrow elongated cylindrical cells which are adjusted to each other, forming a network spanning the entire length of the plant. Interestingly, Zhang et al. [26] demonstrated that the exuding phloem sap of cucurbits is derived from the so-called “extrafascicular” phloem and not the vascular (fascicular) conducting phloem of the vascular bundles. That is, cucurbits have two types of phloem and while the “fascicular” phloem quickly becomes blocked when the stem is cut, the “extrafascicular” phloem loses sap abundantly, perhaps due to the absence of shutter mechanisms (i.e., lack of callose) [26]. In addition to essential nutrients and sugars, the sap of the phloem also contains plant pathogens [27], defensive substances, small molecules such as plant growth substances (i.e., phytohormones), trafficking signals, as well as a diverse population of macromolecules (i.e., certain polypeptides, structured RNAs, small non-coding RNAs, and mRNAs) [28,29,30] the functions of which have yet to be elucidated. However, our understanding of the mechanisms of transport of these molecules into and through the SEs remains incomplete. Interestingly, many of the phytohormones carried by the phloem are involved in systemic defense processes, with jasmonates and salicylic acid being two well-studied examples [31,32].

MYB and NAC (*NAC45* and *NAC86*) transcription factors (TFs) play a central role in differentiation of SE [33,34]. Thus, Altered Phloem Development (*APL*) is a coiled-coil MYB type TF that regulates the SE identity in *Arabidopsis thaliana*, producing nuclear breakdown and reduction of cytoplasmic contents via activation of a family of exonucleases. *NAC45* and *NAC86* are redundant APL-targeted genes and are downstream targets of APL [34] and literature therein. By contrast, genes involved in the initiation of companion cell development and specification have not been determined with certainty thus far. In Section 4, I will refer back to all these TFs because of their implications with regard to the enucleation of SE once its maturation has been reached.

## 3. Proteins in the Heterogeneous Phloem-Sap

The plant phloem is far more complex than originally believed. When participating directly in the transport of photoassimilates from source organs to sinks, phloem responds to stressful infective processes caused by the environment, among other stressors. The stress responses involve a complex metabolic adaptation in which a good number of proteins are directly compromised. However, the mechanism by which certain proteins form a defensive shield against stress is still in need of further investigation. For example, do the proteins move once integrated into the sap stream, or are some of them targeted? If they are, how does the labeling process happen? Unfortunately, this is one of the many unknowns that have yet to be solved when explaining the physiology of the sieve tube. Throughout this section, I will consider the possibility of interaction process between phloematic proteins.

Unlike the xylem vessels, the SE lives over time. The maintenance of the life of any organ involves a considerable expense. Therefore, both the undifferentiated and mature sieve tube must have an adequate metabolic gear. The assembly of the pieces involved in the metabolism of the tube is highly complex. That is why much research is still needed to solve this puzzle carried out in an organ without a nucleus. A great number of pieces of phloem sap from several species (mostly cucurbits) have been examined in detail [22,35]. SE exudates, achieved by several procedures, are the primary source for analyses of sieve tube content. Characterization of the composition of the phloem sap in proteins or other components strongly depends on the collection and sampling methods. The collected sap is then subjected to proteomics, metabolomics, or RNA analysis. In addition to collecting phloem exudates, molecular biotechnologies such as laser-capture microdissection (LCM) and INTACT methods have recently been used to elucidate phloem proteins. Generally, the proteins either maintain the sieve tube flow, protect the SE against pathogens [27], or transmit system-wide signals. Zhang et al. speculate that the “extrafascicular” phloem may be involved in plant defense against pathogens [26]. Having said that, the method of collecting phloem sap must be emphasized when studying the phloem characteristics in detail. It should be noted that organelles are not detectable when collecting phloem sap. This is probably because the SE organelles are anchored through proteins to plasmalemma. This anchor might prevent the organelles from being displaced by the mass flow throughout sieve tube, or alternatively serve to keep the SE organelles close together, in order to facilitate the exchange of compounds via the unstirred layer surrounding the diverse adjacent membranes.

The proteins in the phloem exudate (i.e., more than a thousand) range from 10–200 kDa in size, but the majority of them are within the ranges of 10–40 kDa and 60–70 kDa [36]. Strikingly, the phloem of cucurbits exudes large volumes of sap and has higher concentrations of proteins, up to 100 mg mL^−1^, than the plants of other families (0.1–2.0 mg mL^−1^). Unfortunately, the protein turnover machinery in mature SE is largely unknown. However, a fact has been established: the absence of DNA makes mature SE inaccessible to transcriptomic analysis. That is, we can assume that the presence of proteins inside mature SE is due to its translation occurs prior to the beginning of the SE maturation. Or alternatively, that specific proteins come from neighbor company cells. How companion cells choose determined proteins to export is currently unknown. Unfortunately, it has not been possible thus far to differentiate both populations of proteins (i.e., exportable and not exportable) before the export process to the SE. What is clear is that a large proportion of proteins are enzymes involved in the generation of metabolic precursors, ROS detoxification (e.g., GSTF4 and APX5), proteolysis, synthesis of amino acid (i.e., the second most abundant metabolites in phloem sap after sugars), raffinose metabolism, and physiological maintenance of SE, suggesting that the sieve tubes display high levels of metabolic activity [22,37]. The metabolome and proteome of the phloem sap from two cucurbits was analyzed (i.e., watermelon and cucumber). A specific subset of enzymes has been identified (i.e., 141 and 93 in cucumber and watermelon, respectively). This suggests a specialized metabolism, occurring in the sieve tube system even after differentiation (e.g., enzymes involved in raffinose metabolism). Likewise, some of the proteins specifically localized to the sieve tube are short lived and therefore require continuous *de novo* synthesis. Finding a way to prioritize the passage thorough plasmodesmata of these short-lived proteins with respect to others is another piece of the puzzle that has yet to be resolved. The question is, how is the targeting process carried out in the companion cell, plasmodesmata and SE? On the other hand, it is important to highlight the existence of two ABA transporters in SEs. This finding implies a possible role in systemic ABA transport which is important for acclimation to abiotic stresses [38].

Very recently, Liu et al. (2022) conducted a study using *Nicotiana tabacum* and were able to isolate SEs that were not contaminated with other phloematic cells [39]. This progress provides a valuable tool for exploring the SE biology. Concurrently, this novel protocol allowed the identification of large numbers of highly enriched SE-specific proteins. Prior to this breakthrough, the isolation of sieve tube was very complex, as contamination with adjacent cells was a common occurrence. That is why the sap that emerged from the sieve tube (i.e., phloem exudate) did not faithfully reflect the content of the sieve tube itself [22,35,37,40]. That is, the composition of phloem sap is one of many controversial topics when approaching the study of the sieve tube. Apart from the contaminants (i.e., protein debris caused during maturation process), are there protein components of the sap that are present in all the species studied or, on the contrary, is there a specific protein pattern for each species? We must take into account that the accompanying cells have very high metabolic activity and send a good number of proteins up to a size 70 kDa non-specifically to the SE [41]. It is possible that most of the specifically expressed companion cell proteins are getting lost in the phloem translocation stream mainly driven by their abundance [41] and literature therein. On the other hand, several proteins belonging to the plasmalemma (i.e., integral membrane proteins) of the SE are translated using a short-lived mRNA, for which they require continuous *de novo* synthesis.

In addition to the above, a significant number of ribosomal and proteasomal proteins were also found in a differentiated sieve tube. However, these molecules do not proceed from the companion cells given the size exclusion limit of the plasmodesmata [41,42]. Ribosomal fragments may be remnants from differentiating phloem cells [42], and the proteasomal ones could be involved in defense responses by marking and degrading pathogen proteins [36]. The presence of proteasomes in the cucurbits SE seems possible, but solid evidence of their occurrence and mode of action is lacking thus far. It will be important to demonstrate at what stage in the life of the SE the proteasomal proteins appear. Small proteins (20–70 KDa) from companion cells to SEs may explain the plethora of macromolecules identified in phloem sap [43]. In addition, the majority of molecules found in the phloem stream may simply be the product of macromolecular leakage from companion cells. Taking these last sentences together, the appearance of unexpected metabolic properties (i.e., contradictory features) in the differentiated SE may be a consequence of the variety of approaches used for its study. In other words, protein’s presence in exudates would not necessarily imply a physiological function in these cells. This is the case for the presence of G-actin but no tubulin in the EDTA-facilitated exudate of differentiated SE [22] and literature therein. Recent studies dismiss this fact by attributing the contradictory data to the existence of the remains of the actin-based cytoskeleton once the differentiation is concluded [25]. Thus, the presence of an actin network in SEs is under discussion [44]. Therefore, the occurrence of G-actin in sieve tubes is explained best by the leakage of this ubiquitous protein into transporting tubes from companion cells and differentiating, immature sieve elements. On the other hand, serial block-face scanning electron microscopy has recently been used to study the morphological changes that occur during SE differentiation in greater detail [21,45]. However, among other debated questions related to the knowledge of the sieve tube, it is difficult to accept the existence of an intense flow of compounds between the companion cell and SE in the absence of a cytoskeleton. Hafke et al. (2013) provided evidence of an actin network in mature SEs [23] and literature therein. Nevertheless, contamination of the phloem exudates with degradation products existing in differentiating SE can lead to these contradictory results [23].

Although proteins are the most abundant components of phloem sap, the heterogeneity of other compounds is part of the complexity of the sieve tube integrants (i.e., nucleic, organic and fatty acids, sugars, phytohormones, polyamines, systemic signals, etc.). Very recently, Broussard et al. (2023) compiled a detailed bibliographic list of sap metabolites via metabolomics from different species based on different sap sampling methods, and distinct analytical techniques [22] and literature therein. On the other hand, the structural proteins (dispersive and non-dispersive P-proteins) and soluble P-proteins are two forms of P-proteins, both of which are found in phloem exudates at high levels. Little is known about the soluble protein content of SEs [22]. It is interesting to reflect that certain proteins are not found in pure phloem exudates. This is the case for the contractile proteins (SEO) that are part of forisomes, and non-contractile (SEOR) ones. Both populations of proteins could be retained inside SE. This is one of many peculiarities of the SE that has yet to be clarified. Aoki et al. (2005) have conducted a novel experiment. To do this, phloem proteins from pumpkin (*Cucurbita maxima)* were used to display the capacity of specific proteins to move long-distance in rice (*Orza sativa)* sieve tubes. They demonstrate that in addition to a passive movement of certain proteins from pumpkin *Cucurbita maxima*, long-distance movement of other proteins is a controlled process and that protein destination is regulated via protein–protein interaction [46]. This is the case for the proteins CmPP16-1 and CmPP16-2 which interact with common phloem sap proteins under natural conditions [46]. That is, phloem proteins move long distances through a combination of two modes: non-selective transport via bulk flow, and selective regulated movement [46]. The PP1 from pumpkin *Cucurbita maxima* was the first structural P-protein (96 KDa) to be characterized. Interestingly, this filamentous protein was immunolocalized in SE slime plugs and P-protein bodies, whereas the corresponding mRNA was shown to accumulate in companion cells [47]. Curiously, no P-protein bodies were observed in the Arabidopsis companion cell [22]. Comparative proteomics in a sieve tube identified fluctuations in protein composition under various environmental conditions, providing further evidence of controlled trafficking [48]. This is the case for FT-INTERACTING PROTEIN 1 (FTIP1), an essential regulator synthesized both in young SEs and in companion cells which is required for *FLOWERING-T* (FT) protein transport in *Arabidopsis thaliana,* demonstrating that FT moves in phloem in a regulated manner [49,50]. Another interesting example of protein interrelation inside a sieve tube is the PHLOEM LIPID ASSOCIATED FAMILY PROTEIN (PLAFP)–PHOSPHOLIPASE Dα1-derived Phosphatidic acid (PtdOH). PtdOH is a known stress signal involved in ABA signaling and detected in phloem exudate. The PLAFP–PtdOH complex constitutes a possible phloem-mobile signal for the systemic coordination of drought response [51].

Differentiating angiosperm-nucleated SEs produces abundant phloem-specific proteins (P-proteins) before their protein synthesis machinery is degraded [23] and literature therein. In young SEs, P-proteins form dense bodies in cytoplasm. However, when the SE acquires maturation and selective autolysis is triggered, P-proteins disperse into individual filaments which move to the periphery of the cell, remaining in this position in the mature functional sieve tube. Remarkably, P-proteins have been described as granular, fibrillar, or tubular, even within the same cell. This may represent the various different stages of P-protein differentiation. In some cases, the dense protein agglomerations remain intact and are visible in functional sieve tubes as non-dispersive P-protein bodies observed in about 10% of Angiosperm families [23] and literature therein. However, whether *P-proteins* represent a family of proteins with similar primary structure remains uncertain. The plugging role of P-proteins in the pores of the sieve-plate to facilitate rapid wound sealing after injury is a possibility that is currently being considered [52]. To achieve this, and following injury, conventional P-proteins detach from their parietal position and form plugs at sieve plates to block further translocation. Indeed, the pores of damaged sieve tubes become entirely filled with protein fibrillar material. Nevertheless, some specific and mobile sieve tube proteins are visible as transcellular strands in all dicotyledonous and most monocotyledonous tubes. PHLOEM PROTEIN-2 (PP2) is a dimeric chitin-binding lectin that is mobilized by SEs once it has been synthetized by the companion cells [53]. Beneteau et al. (2010) suggested that AtPP2-A1 has a dual function: it is involved in the trafficking of endogenous proteins and in the interactions with phloem-feeding insects [54]. Likewise, in cucurbits, PP2s display additional properties, including RNA binding, among other functions. The companion cells also translates the SUT1-mRNA. However, this protein (i.e., sucrose transporter) is localized in plasmalemma of SE [55]. Recently, it was revealed that the subcellular dynamics of these SUT transporters were part of complex regulatory mechanisms. In other words, it has been concluded that (transient) protein–protein interactions of integral membrane proteins help to sequester SUTs to subcellular compartments, such as membrane microdomains, with specific functions to enable subcellular transport and cell-to-cell trafficking via plasmodesmata [56] and literature therein. Interestingly, Marco et al. (2021) showed that a phytoplasma infection compromises the function of *SUT1* and *SUT2* [57]; *whereas* van Bel (2021) updates the location in plasmalemma both in monocots and eudicots of specific sucrose transporters SUTs, STPs, SUCs, and SWEETs [58] and literature therein.

Finally, Azizpor et al. (2022) carried out a striking immunological experiment using the reciprocal oligosaccharide probe OGA^488^ [59]. This probe created reproducible and highly specific staining of P-protein bodies in poplar, bean and cucumber stems. The result was unexpected, as the OGA^488^ probe was designed to reciprocally bind to chitin, and plant cells do not make chitin. Using previously characterized *LOF* mutants for Arabidopsis, P-protein components *AtSEOR1* and *AtSEOR2*, we found that AtSEOR2 is responsible for binding OGA^488^. The Azizpor results provide a new method for rapidly labeling P-protein bodies in sectioned material and raise questions about the significance of carbohydrate binding in the function of P-protein bodies. On the other hand, since these results come from stem sections, they probably do not reflect the native form of P-protein bodies in unwounded tissues [59]. These results failed to label P-proteins from unsectioned *Arabidopsis thaliana* roots. OGA^488^ is thus robust and easily used to label P-proteins in histological sections of multiple angiosperm species. Its use in intact material needs to be investigated in depth.

To summarize, for the data we have, it is evident that heterogeneous and mobile proteins, which are crucial for tight plant development and stress response, are purposely transported inside the sieve tube. This assumes that the phloem has a control system for the translocation of signal proteins. In this system, the molecular interrelation must be key. As previously discussed, who makes the decision in the companion cell for the choice of a specific protein for a given answer in SE remains a mystery. Together, although SE and the companion cell seem to constitute a functional unit, the companion cell specification and the mechanism of dependence between the two cells is in need of further investigation. As previously discussed, who makes the decision in the companion cell to choose a specific protein for a given answer in SE is a mystery. If some are located on the opposite side, what is the mechanism they adopt to cross the sap stream from the companion cell? Is it affected by the presence or absence of light? Likewise, companion cells deliver enzymatic proteins needed for SE (e.g., several enzymes related to protein degradation). However, it has yet to be demonstrated how enzymes are produced by the companion cell and trafficked to the SE. That is, who orders their synthesis, in what form they are transported, and what is the temporality of their translation? These and many other questions currently underlie the way phloem works.

Lipophilic compounds are not habitually expected in sieve tube exudates because the sap is an aqueous hydrophilic environment. Therefore, while some molecules may act in a sieve tube as independent mobile signals, lipophilic compounds must interact with proteins to facilitate their transport and signaling activity. This type of transport has been little studied in plants, and not much data exist regarding the transport of lipophilic molecules inside sieve tubes [38]. Some lipids may not only be involved in intracellular signaling but may also play an important role in long-distance signaling. Lipid-binding proteins (LBPs) function as chaperones (e.g., annexin [60]) which move lipids involving Ca^+2^ in the process. The study of phloematic chaperones is not a novelty. Let us consider an example that has been identified in a diverse range of works and, interestingly, was suppressed in the phloem of *Arabidopsis thaliana* during infection with the pathogen *Pseudomonas syringae* [27,48] and literature therein The following lipophilic molecules are found into phloem sap: (i) hormones such as free-ABA, phaseic and dihydrophaseic acids, GAs (gibberellin A1), IAA (glycosyl-ester; cell to cell in carrier-mediated transport), CKs [N-6-(2-isopentenyl) adenine-type cytokinin-ribosides] [61], SA (salicylic acid, methyl salicylate), and JA/oxylipins (methyljasmonate, jasmonate isoleucine) [13,62] and literature therein. While the lipophilic hormones are structurally different, they appear to move through the sieve tube either in their free form or as conjugates with amino acids, methyl groups, or sugar (glycosyl/ribosyl) groups. Carrier proteins do not appear to be necessary for the long-distance movement [38] and literature therein of (ii) fatty acids and (iii) glycerolipids such as diacylglyerol [DAG; probably bound to phloem-localized LBPs], triacylglycerol (TAG) (LBP-lipid complex?), phosphatidyl choline (PC) (LBP-lipid complex?), phosphatidyl inositol (PI) (LBP-lipid complex?), and phosphatidic acid (PA) (LBP-lipid complex?) [63] and literature therein. Recently, phospholipids have emerged as possible long-distance signals as well [64]. Due to their hydrophobic nature, it was proposed that phloem LBPs participate in different aspects of this lipophilic signaling cascade [65] and literature therein. Hoffmann-Benning’s group suggests that to participate in the proposed long-distance lipid signaling, LBPs must (i) bind to a specific lipid also present in the phloem sap; (ii) their genes need to be specifically active in the companion cells; (iii) LBPs must be able to reach the SE through plasmodesmata; and (iv) the expression of the gene encoding LPB is likely controlled by the same factors as the production of the lipid–ligand [38] and literature therein. It is feasible to hypothesize that, with the probable exception of (iv), the signaling of non-lipophilic compounds must possess the first three suggested features.

In order to continue the debate on the existence of heterogeneous proteins in the differentiated SE, and try to find coherent explanations for this, Garg and Kühn (2020) ask themselves a very interesting question: what determines the composition of the phloem sap? Their main arguments are as follows: (i) the existence of short-lived proteins that are exclusively present in SE strongly argues for an undocumented but specific transport of macromolecules through pore plasmodesma units (PPUs); (ii) the presence of active proteosome (unquestionable) causes a different degree of contamination during the SE differentiation process, and this degradative process does not seem to be comparable along the sieve tube; (iii) the demand for proteins is not identical in the SE along its differentiation phases and the activity of protein recycling, either. Likewise, the SE protein requirement also depends on the section of the sieve tube to be considered [66]. Together, these and other conjectures have an indisputable point of coincidence. That is, there is no doubt that the vast majority of questions and approaches about the heterogeneity of phloem sap will be clarified once the morphology and physiology of the plasmodesmata that connects the companion cell and SE have been studied in depth. Thus, some plasmodesmata proteins are able to interact with certain sucrose transporters [55].

## 4. The Translational Machinery in Differentiated SE Is Disrupted by Enucleation

During the final stage of SE differentiation, the nuclear disintegration process (enucleation) takes place, along with the loss of some additional key organelles (e.g., functional ribosomes required for translation have not been observed in mature SE) [23] and literature therein. Therefore, the SE becomes a cell that is metabolically dependent on associated nucleate companion cells with which SE is connected by plasmodesmata [67] and literature therein (Figure 1). Before starting enucleation, the nuclear membrane becomes disorganized and the nucleus decreases in size. Likewise, small lytic vacuoles were shown to endure before and after enucleation. The finding that companion cells are not involved in phloem unloading from SEs raises interesting questions regarding molecular trafficking across plasmodesmata. For example, do the PPUs remain open, supporting continuous molecular movement from companion cells to SEs? Or, on the contrary, are they only transiently open depending on the state of SE maturation?

Arabidopsis NAC45/86-DEPENDENT EXONUCLEASE-DOMAIN PROTEINS (NENs) direct SE morphogenesis culminating in enucleation [21]. *NAC45/86* encodes a TF that regulates the downstream processes of SE maturation, whereas *NEN1* and *NEN4* encode exonucleases responsible for the degradation of nuclei in SE. The organization of SE is controlled by: (i) the OCTOPUS–BRASSINOSTEROID INSENSITIVE 2-BRI1-EMS-SUPPRESSOR 1 (OPS-BIN2-BES1) regulatory pathway which controls SE differentiation; and (ii) the ALTERED PHLOEM DEVELOPMENT (APL)-NAC45/86-NEN regulatory pathway which controls SE enucleation. OCTOPUS is a polarly localized membrane-associated protein that (*a*) regulates phloem differentiation entry in *Arabidopsis thaliana*. [68]; and (*b*) negatively regulates BIN2 to control phloem differentiation in *Arabidopsis thaliana* [69]. However, the existence of a crosstalk between OPS-BIN2-BES1 and APL-NAC45/86-NEN regulatory pathways has yet to be proven. This fact constitutes one of many weaknesses in the experimentation of the phloem. During enucleation, which lasts 10 min, the rupture of the nuclear membrane and nucleole disintegration take place, followed by the expulsion of the nuclear content for its subsequent degradation in the parietal cytoplasm (i.e., mictoplasm) [21]. Despite the recent identification of APL-dependent exonucleases executing nucleus degradation and cytosol dilution, the molecular mechanisms regulating CW thickening, sieve plate formation, and cytoskeleton rearrangement remain poorly understood. Interestingly, after enucleation, APL is detected only in companion cells and metaphloem [21,67].

The enucleation is one of the main processes carried out during differentiation of SE, disrupting the protein synthesis machinery in a differentiated sieve tube [42]. However, it was suggested that at least some components of the protein synthesis machinery are present in pumpkin phloem sap, raising the possibility that limited translation may occur in mature SE [37]. But this translational process is difficult to prove and explain in view of the remarkable current knowledge on the protein synthesis in eukaryote organisms. The use of isolated and functional SEs [39] would shed light on these faltering possibilities. Sanden and Schulz (2021) present interesting data suggesting that young SEs translate part of the protein machinery found in mature SEs [55]. Interestingly, these authors describe highly specific proteins that localize to mature SEs without entering the phloem translocation stream (e.g., H^+^-ATPase, sucrose transporter, aquaporin, lectin, SE occlusion and SEO related, Ca^2+^, FL-T chaperone, β- amylase), and callose synthase [55]. Do all vascular plants share one set of sieve element proteins? This is an interesting open question, what determines if a specific protein is degraded in the enucleation process. Alternatively, is sieve element specificity a dynamic feature that may be toggled on and off for specific genes during evolution? In [70], Sanden and Schulz (2022) selected several genes and discuss the potential roles of codified proteins in mature SEs. At present, it seems beyond doubt that physiological functions of the viable mature enucleate SE are mainly maintained by nucleate companion cells (i.e., specialized parenchyma cells with highly active mitochondria, high phosphatase activity, and an absence of starch) [13,71]. As said previously, ER displays acid phosphatase activity, suggesting it plays a role in the cytoplasmic autolysis during SE maturation [23].

Unlike SEs, companion cells are characterized by their increased cellular density (i.e., large nucleus and numerous ribosomes in ER). That said, and given their neighborhood, companion cells are supposed to help maintain the viability of the SEs. Nevertheless, the likelihood that more help is provided by the companion cells is far from being determined. The plasmodesmic traffic from companion cell to SE is very complex and mostly unexplored, just considering the four accessible routes: the plasma membrane, the cytoplasmic sleeve, the desmotubule membrane, and the desmotubule lumen. The presence of a good number of proteins in different routes does not rule out the possibility that some passing proteins become bound, making transit difficult. It has not been ruled out that companion cells transfer energy and specific proteins to enucleate SEs [72]. We must not forget that SEs exhibit important metabolic activity. The potential roles of proteins in mature SEs were recently reviewed [66]. As the authors of the review point out, it will be interesting to determine the physiological significance of the spatial separation of the proteins within SE plasmalemma. Similarly, genes involved in the initiation of companion cell development have not been determined with certainty thus far [43]. SE/companion cell complexes arise from (pro)cambial mother cells induced by key genes known to be decisive for SE differentiation [24]. It was long thought that SEs and companion cells originate from a common precursor cell. However, at least in the Arabidopsis roots, they come from different procambium cells [25,34]. *DOF* TFs preferentially expressed in the phloem (phloem-*Dof*s) are not only necessary and sufficient for SE and companion cell differentiation. DOF not only induces the *OCTOPUS* positive regulator, but also negative regulators (e.g., CLAVATA3/EMBRYO SURROUNDING REGION RELATED) (*CLE*) of phloem development [73,74]. In summary, the Dof-CLE circuit controls phloem organization. In 2023, the *Arabidopsis thaliana CLE33* gene was identified and demonstrated to be an essential player in protophloem formation [75].

## 5. A Long Way to Go

This publication updates the current state of knowledge on the SE/companion cell relationship. Throughout the paper, it is clearly observed that a large amount of research is required to understand the functioning of this physiological unit, which will lead to the development of tools ensuring that the processes that operate in the sieve tube are developed properly.

These are some goals to carry out:The mitochondria of the SE are less efficient than those of the companion cell—what is their function and what is the extent of their collaboration in the sieve tube?Is there targeting in some phloem sap proteins? If so, how is this process carried out in plasmodesmata, plasmalemma, and companion cell?Do all vascular plants share one set of SE proteins?Considering the existence of sap flow, how does the movement of proteins from the companion cell take place to wade through the phloem current and access the plasmalemma located on the opposite side of the companion cell? Does light affect this movement?How is the companion cell protein synthesis ordered, and by whom? In what form are the proteins transported, and what is the temporality of their translation?What is the function of LBP proteins in the sieve tube?Who determines the composition of the phloem sap?Do the PPUs remain continually open? Or are they only transiently open depending on the state of SE maturation? Who determines the degree of openness in mature SE?How did evolution affect the characteristics of the sieve tube in Angiosperms?The cucurbits are unique species in evolving dual phloem transport systems?The occurrence of two companion cells is reported to be associated with a single SE in some species. But how can both companion cells coordinate with a single SE and also control the functions of SE?

## Figures and Tables

**Figure 1 plants-12-03033-f001:**
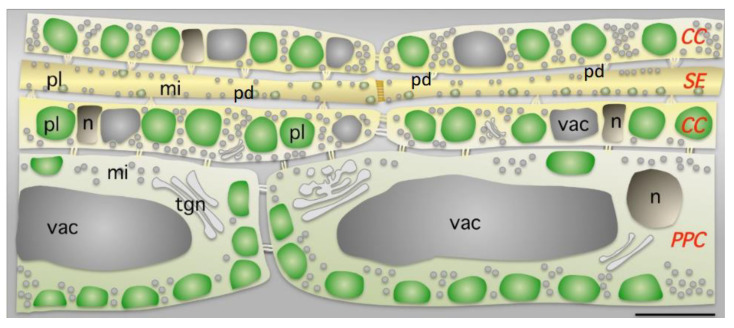
Parenchyma phloematic cell (PPC), sieve element (SE), and companion cell (CC), and distribution of mitochondria (mi), nucleus (n) and plastids (pl) in photosynthetic leaves. In the CCs, plastids are aligned and occupy a large proportion of the cell volume. Numerous mitochondria (mi) are also found in each cell type. The nuclei (nu) and the vacuole (vac) have unusual shapes and positions in the CCs. Plastid size differs with cell type (1 μm in SE, 3 μm in CC and 4 μm in PPC), whereas mitochondrial size variations are more limited (0.6 μm in SE, 0.6 μm in CC and 0.7 μm in PPC). Numerous plasmodesmata (pd) exist in the SE/companion cell physiological unit. This figure is reproduced from [20].

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
