# Peer review of "The Interplay between Enucleated Sieve Elements and Companion Cells"

_plants, 2023, doi:10.3390/plants12173033_

Round 1

Reviewer 1 Report

I have gone through the manuscript submitted by Dr. Angel J. Mattila, which is well written and data compilation is satisfactory. However, there are some terminological issues (e.g., vessel tube, word “tube” is always used along with sieve elements and not with vessels) that needs to be addressed. I have highlighted some of the sentences on manuscript directly (pl. see annotated MS).  

However, I feel there is a scope for the improvement of the manuscript because several times Cucurbits sap is mentioned but some of important papers are not cited (pl. note that I am not author in any of these papers; listed below) that provides valuable information on the cucurbit phloem. I feel it will provide an important input and improve the existing MS.

Another important point is Cucurbitaceae is well known for the occurrence of extra-fascicular phloem, which is completely lacking. Similarly, the occurrence of two companion cells is reported to be associated with a single sieve element in some species. Since, it is a review article, if some light is thrown on this aspect that how two companion cells coordinate with single sieve element and how they control the function of sieve element in such scenario.

Recommended articles:

Oparka K.J. & Cruz S.S. 2000. The great escape: Phloem transport and uploading of macromolecules. Annu. Rev. Plant Physiol. Plant Mol. Biol. 2000. 51:323–47

Turgeon, R., Oparka, K., 2010. The secret phloem of pumpkins. Proc. Natl. Acad. Sci. (USA) 107, 13201–13202. https://doi.org/10.1073/pnas.1008134107  

Zhang, B., Tolstikov, V., Turnbull, C., Hicks, L.M., Fiehn, O., 2010. Divergent metabolome and proteome suggest functional independence of dual phloem transport systems in cucurbits. Proc. Natl. Acad. Sci. (USA) 107, 13532–13537. https://doi.org/10.1073/pnas.0910558107

Azizpor, P., Sullivan, L., Lim, A., Groover, A., 2022. Facile labelling of sieve element phloem-protein bodies using the reciprocal oligosaccharide probe OGA488. Frontiers in Plant Sciences 13:809923. https://doi.org/10.3389/fpls.2022.809923

I also suggest to refer the chapter on phloem by Evert R.F. 2006. Esau’s Plant anatomy.

I recommend the manuscript for its acceptance in your esteemed journal.

Author Response

Dear Ref. 1,

  • your reccomendations have been of great help to further update this review on phloem.
  • the first recommendation was included as suggest. 
  • due to its important scientific contribution in the field referred in this update, all reference suggested by you have been included and commented into new text. 
  • finally, your last very interesting comment was taken into account by including in text.
  • Best regards 

Reviewer 2 Report

The manuscript summarized the current understanding of the role of protein components found in the phloem and the development of phloem tissue with emphasis on the enucleation process during differentiation of the distinct cell type sieve-element. The manuscript is well written with elegant science narrative and clarity. It is acceptable for publication in the renowned scientific journal PLANTS.

However, the title does not fit with the manuscript content, it should be more suitable. Fig 1 and 2 are oversimplified, one more scheme that illustrates the positions of xylem and phloem tissues and summaries those important factors/genes being found (mentioned in the manuscript) in the phloem and regulate the differentiation should be presented. In addition to collecting phloem exudates, molecular biotechnologies such as laser-capture microdissection (LCM) and INTACT methods to elucidate phloem proteins are worth mentioning in section 3.

Minor comments:

- Figure 1, plasmodesmata exist in the SE/CC should be indicated in the figure.

- After a species name introduced in the text, the generic name is abbreviated in subsequent mentions.

- Many terms were unnecessarily italicized.

- Page 6 of 14, check the binominal name (the first part) of rice plant.

- The numerical and author-date citation styles were mixedly used in the text.

- Full name and abbreviation were randomly used making it rather difficult to follow.

Author Response

Dear Referee 2,

  I appreciate your suggestions which have served to eliminate some errors and add news in my update. I've taken into account all your suggestions.

    Best regards